# Hepatic Metastasectomy in Squamous Cell Carcinoma of the Anal Canal: A Case Series of a Curative Approach

**DOI:** 10.3390/cancers15153890

**Published:** 2023-07-31

**Authors:** Jane E. Rogers, Amanda Sirisaengtaksin, Michael Leung, Van K. Morris, Lianchun Xiao, Ryan Huey, Robert Wolff, Cathy Eng, Jean Nicolas Vauthey, Ching-Wei D. Tzeng, Benny Johnson

**Affiliations:** 1Pharmacy Clinical Programs, U.T. M.D. Anderson Cancer Center, Houston, TX 77030, USA; 2Department of Gastrointestinal Medical Oncology, U.T. M.D. Anderson Cancer Center, 1515 Holcombe Blvd, Houston, TX 77030, USA; 3Department of Biostatistics, U.T. M.D. Anderson Cancer Center, Houston, TX 77030, USA; 4Vanderbilt Department of Medical Oncology, Nashville, TN 37232, USA; 5Department of Surgical Oncology, U.T. M.D. Anderson Cancer Center, Houston, TX 77030, USA

**Keywords:** anal neoplasms, hepatectomy, oligometastatic

## Abstract

**Simple Summary:**

Our review summarizes the experience of eight hepatic-only metastatic squamous cell carcinoma of the anal canal cases who underwent hepatic resection. Metastatic squamous cell carcinoma of the anal canal patients are traditionally treated systemically with carboplatin plus paclitaxel front-line followed by single agent programmed-death-1 inhibitors. Following these two regimens, systemic options are limited. Our patients reported here underwent surgical resection after neoadjuvant systemic therapy. Outcomes were favorable revealing optimism with a multidisciplinary approach.

**Abstract:**

Background: Squamous cell carcinoma of the anal canal (SCCA) is rare. Most cases are diagnosed in a localized setting. Metastatic SCCA is rare, and investigation has been limited in the past for these patients. We believe that hepatic-only metastatic disease could have a unique treatment landscape compared to diseases with diffuse metastatic involvement. Here, we describe cases at our institution. Methods: We reviewed eight SCCA cases with hepatic-only metastatic disease (diagnosed February 2018–January 2022). The objectives were to determine the overall survival and disease-free survival with this approach. Results: The median age was 62 years old (yo). Patients had an ECOG of 0–1. All patients received definitive chemoradiation to their primary anal tumor. A median of three months of neoadjuvant systemic therapy was provided. All patients had a response on their first scan after systemic therapy. Sixty-two percent received carboplatin + paclitaxel. A complete pathologic response was seen in 62% of patients. At their last follow-up, all patients were alive. Three patients had recurrent disease. The estimated 1-year disease-free survival probability was 56.2%. Conclusion: Our report shows the feasibility of a curative-intent approach for patients with hepatic-only metastatic SCCA following the neoadjuvant application of carboplatin + paclitaxel. This approach appears promising in these select patients and warrants further investigation.

## 1. Introduction

Anal neoplasms account for ~2.7% of all digestive system cancers, with 9440 estimated new cases in 2022 [1]. Squamous cell carcinoma of the anal canal (SCCA) is the most common type of anal neoplasm [2]. SCCA is a human papillomavirus (HPV)-associated malignancy with ~90% association. Despite the availability of HPV vaccination, compliance with the vaccination series continues to be an issue [3]. This lack of vaccination compliance leads to SCCA persisting as a continued healthcare burden.

Most SCCA patients present with localized or locally advanced regional disease, which has a high cure rate with definitive chemoradiation (CRT) [2]. Salvaging abdominoperineal resection (APR) is reserved for localized patients who fail definitive CRT. Metastatic disease is rare, occurring in 10–20% of patients and most often as metachronous recurrence after primary localized treatment. The liver represents the most common site of metastases outside the pelvis and the inguinal nodal basins. Historically, due to the rarity of metastatic SCCA, existing prospective data have been limited. Treatment recommendations and outcome data have often been based on retrospective reports, case series, and/or extrapolating treatments from other squamous cell malignancies. 

The INTERnational Advanced Anal Cancer (InterAACT) study established a standard front-line therapy for the management of patients with metastatic SCCA. InterAACT was a phase II front-line chemotherapy trial that aimed to determine the most efficacious and tolerable front-line platinum-based chemotherapy regimen [4]. Metastatic SCCA patients were randomized to cisplatin at 60 mg/m^2^ intravenous (IV) + 5-Fluorouracil (5-FU) 1000 mg/m^2^/day continuous IV infusion on days 1–4 every three weeks or to carboplatin AUC = 5 IV day 1 + paclitaxel 80 mg/m^2^ IV on days 1, 8, and 15 every four weeks. Response rates were similar between both groups (57.1% with cisplatin + 5-FU vs. 59% with carboplatin + paclitaxel). Median progression-free survival (PFS) was not statistically different (cisplatin + 5-FU median PFS was 5.7 months vs. carboplatin + paclitaxel median PFS was 8.1 months, *p* = 0.375). Median overall survival (OS) was statistically different, favoring the carboplatin + paclitaxel arm (cisplatin + 5-FU median OS was 12.3 months vs. carboplatin + paclitaxel median OS was 20 months, *p* = 0.014). Additionally, a more favorable toxicity profile was seen with carboplatin + paclitaxel. Following these pivotal data from InterAACT, carboplatin + paclitaxel has become the preferred regimen of choice for metastatic SCCA [2]. Of note, 71% (n = 65) of patients had two or more sites of metastatic disease in InterAACT [4]. 

Multidisciplinary approaches have been described retrospectively and through post hoc exploration [5,6,7,8]. A standard approach for those with oligometastatic disease is still lacking. Here, we review our institutional experience with eight patients treated with systemic therapy followed by hepatectomy with a curative intent to highlight such an approach as potentially meaningful for patient outcomes. These patients additionally had definitive CRT to the primary anal tumor. 

## 2. Materials and Methods

We performed a retrospective evaluation of eight patients treated at our center. Patients were diagnosed with metastatic disease between February 2018 and January 2022. Patients had hepatic-only metastatic SCCA disease. Points for data collection included baseline demographics (age, gender, race), the presence of human immunodeficiency virus/acquired immunodeficiency syndrome (HIV/AIDS) along with baseline CD4 count at chemotherapy start, any medications for HIV drug–chemotherapy interactions, the known presence of HPV, Eastern Cooperative Oncology Group (ECOG) performance status, histology grade, and prior treatment history. A mutation profile was reported when available. Reported outcomes included post-diagnosis OS (metastatic disease diagnosis date to death date), disease-free survival (DFS) (hepatectomy surgical date to recurrence date), recurrence rate in all patients, pathologic response to treatment, toxicities requiring a delay or a dose adjustment, and best response (any shrinkage, stable disease, or progression) with systemic therapy. 

Summary statistics were used to describe the study population. Categorical variables were tabulated with frequency and percentage, and continuous variables were summarized using descriptive statistics. The Kaplan–Meier method was used to estimate median follow-up time for all patients and the probability of overall and disease-free survival.

## 3. Results

Baseline demographics are included in Table 1. The median age of our eight patients was 62 years old (range 43–74). Fifty percent of patients had metastatic disease at diagnosis, while the remaining fifty percent had metachronous disease. All patients were Caucasian, and most were female (n = 7). All patients had a poorly differentiated histology. Three patients had mutational profiles performed, and two of these patients had *PIK3CA* mutations. No patients were HIV-positive. HPV was not consistently collected. All patients had an ECOG performance status of 0–1 at the start of systemic treatment for metastatic disease. All patients underwent hepatic resection following front-line systemic therapy. Definitive CRT to the primary anal tumor was given to all patients, with the majority before liver resection (n = 7). Chemotherapy regimen given for definitive CRT was divided 50:50 into 5-FU + cisplatin and 5-FU + mitomycin C. The median duration of metastatic front-line systemic therapy provided prior to surgical resection was 3 months (range 2–5 months). All patients had a response based on imaging prior to surgical resection. Carboplatin + paclitaxel was given in 62% of patients (n = 5), with carboplatin dosed at AUC = 5 and paclitaxel at 175 mg/m^2^ every 21 days. One patient had carboplatin AUC = 2 + paclitaxel 65 mg/m^2^ given weekly. Cisplatin at 40 mg/m^2^ + 5-FU 2400 mg/m^2^ continuous infusion over 46 h every 14 days was given in two patients (25%). One patient was enrolled on a front-line clinical trial. Most patients had a delay in treatment due to toxicity (75%). Myelosuppression and fatigue were the most common reasons for a delay in treatment.

The median follow-up time was 18.9 months (95% CI: 17.4 months-NA) for our cohort of patients. Five patients (62%) had a pathologic complete response in their liver tissue following resection. One patient had a moderate treatment response (5% tumor viability), one patient had a variable tumor response (40–100% viability), and one patient had no definitive treatment response (90% tumor viability). Of note, the patient with 40–100% tumor viability had an *FBXW7* mutation (as well as mutations in *PIK3CA* and *KMT2D*). The patient with 90% tumor viability had mutations in *DCLRE1C*, *NOTCH1*, and *TP53*. Three patients showed recurrence following surgical resection. The sites of recurrence were the liver (n = 2) and the hepatic artery lymphadenopathy (n = 1). The case with recurrence in the hepatic artery lymphadenopathy was treated with radiation followed by surgical resection, and the patient remained without disease at the last follow-up (3 months post-resection). The other two cases with liver recurrence were treated with systemic therapy +/− ablation to these lesions. Four of the five patients with complete pathologic response have yet to show recurrence based on the last follow-up. The additional patient without recurrence had a moderate treatment response in their pathology (5% tumor viability). Two of the patients who had recurrence had higher tumor viability in their pathology (40–100% viability and 90% viability). No patients have died to date, and the estimated one-year disease-free probability was 56.2% (95% CI 28.1–100%) (Figure 1). The specific outcomes for each case are described in Table 2. Two patients have been disease-free for more than 4 years. 

## 4. Discussion

Our review shows the clinical feasibility of providing a curative approach for patients with hepatic-only metastatic SCCA. Five of our patients (62%) were free of disease for greater than 1 year, and two patients have continued to be disease-free for over 4 years since hepatectomy. Traditional treatment for metastatic SCCA is with continued systemic doublet chemotherapy (carboplatin + paclitaxel) until progression as a result of the phase 2 InterAAct trial. Single-agent immune-checkpoint therapy (nivolumab or pembrolizumab) is given as a second-line therapy upon progression. These single-agent anti-programmed-death-1 (anti-PD-1) agents, nivolumab and pembrolizumab, are recommended based on the phase 2 NCI9673 trial and the phase 1b KEYNOTE-028 [9,10]. The NCI9673 was a multicenter, single-arm study of nivolumab [9]. Patients (n = 37) were treatment-refractory with a median of two prior therapies (range 1–7). The overall response rate was 24% (n = 9), with a median duration of response of 5.8 months. The median PFS was 4.1 months, with a median OS of 11.5 months. The median follow-up time was 10.1 months. At the time of data cut-off, 65% of patients had progressed off of treatment. KEYNOTE-028 was a multicenter, single-arm study of pembrolizumab in treatment-refractory patients (n = 43) (52% with ≥2 lines of therapy) [10]. The overall response rate was 17%. The median duration of response was not reached (range < 0.1 + month-9.2+ month). The median PFS was 3 months, the 6-month PFS was 31.6%, the 12-month PFS was 19.7%, and the median OS was 9.3 months. Similar results were seen in the POD1UM-202 study evaluating retifanlimab, an anti-PD-1 antibody [11]. As these results show, there is clear intrinsic and acquired resistance with single-agent anti-PD-1 agents in refractory SCCA as not all patients show a response and the duration of response still needs improvement. Investigations are underway for ways to overcome these resistance pathways and examining immunotherapy in early-stage disease [12,13,14,15,16,17,18,19,20,21,22,23].

Following second-line immune-checkpoint therapy, other platinum doublets (cisplatin + 5-FU or fluoropyrmidine + oxaliplatin) and/or single-agent anti-epidermal growth factor receptor (EGFR) antibodies) represent potential remaining options utilized based on limited experience and responses seen in early-stage disease [2,24,25,26,27,28,29,30]. These systemic regimens, however, come with limitations and unknowns. Platinum combinations following the failure of carboplatin plus paclitaxel raise some concern about potential efficacy. Anti-EGFR therapy in the metastatic setting, despite limited RAS mutations in SCCA, has shown limited efficacy. The phase 2 trial of avelumab, an anti-programmed death ligand-1 antibody, in combination with cetuximab, an anti-EGFR antibody, showed that for the combination, the overall response rate was 17% with a median PFS of 3.9 months and a median OS of 7.8 months [26]. Kim et al. explored anti-EGFR therapy alone or in combination with irinotecan and showed ~30% response, a median PFS of 4.4 months, and a median OS of 11.4 months [25]. Our report on 56 patients given anti-EGFR therapy plus or minus a variety of chemotherapy regimens showed a response of 41%, a median PFS of 4.3 months, and a median OS of 16 months [24]. Therefore, given the limited efficacy and data with these refractory agents, patients are often referred for early-phase clinical trials. Other targets are under investigation with hopes of improving the arsenal of agents available for this population [31].

As described above, there is not a robust amount of systemic therapy options in metastatic SCCA, and therefore, other therapies should be considered. Additionally, systemic therapy for metastatic SCCA is not without adverse effects, as seen by 75% of our patients requiring a delay in care due to underlying treatment-related toxicity. Our approach (neoadjuvant systemic therapy followed by hepatectomy) allowed for time off chemotherapy as patients were not recommended to receive “adjuvant” treatment, and in some cases this approach was curative. *PIK3CA* was present in 25% of patients, which is consistent with the literature, showing a potential targeted therapy to be explored [32]. Most notably, a pathologic complete response (pCR) was achieved in 62% of our patients, confirming activity with platinum doublet therapy. Additionally, one patient with a pCR received a clinical trial with immune-checkpoint inhibition upfront, showing the potential promise of a neoadjuvant immunotherapy exploration in oligometastatic patients. Patients without recurrence showed a high pathological response (n = 4 had pCR; n = 1 had 5% tumor viability). Two of the three patients with recurrence had a high tumor viability after systemic therapy. One patient with recurrence had an *FBXW7* mutation, which appears to hold clinical significance in tumor development, progression, and treatment resistance [33]. 

Multidisciplinary care for Metastatic SCCA has previously been described in the literature [5,6,7,8]. Eng et al. reported retrospective outcomes at our institution of metastatic SCCA patients (January 2000–May 2012) [5]. Patients (n = 33) who underwent multidisciplinary care (surgery; CRT) had improved outcomes compared to those given palliative systemic chemotherapy alone (median OS 53 months vs. 17 months, *p* < 0.001; median PFS of 16 months vs. 5 months, *p* < 0.001). Multidisciplinary care consisted of surgery (n = 19) and chemoradiation (n = 14). The types of surgical treatment provided included liver resection (n = 9), lung resection (n = 2), lymph node dissection (n = 5), radical pelvic resection (n = 4), and radiofrequency ablation (n = 3). Kim et al. reported a post hoc analysis of the Epitopes-HPV02 phase 2 study (September 2014–December 2016) [6]. Epitopes-HPV02 was a single-arm study evaluating docetaxel + cisplatin + 5-FU (DCF) front-line for metastatic SCCA patients. Complementary local treatments (surgery; CRT) were allowed following the end of planned DCF cycles. Twenty-one percent (n = 14) underwent these approaches. Twelve percent (n = 8) underwent surgery for their metastatic disease. A pathologic complete response was seen in 63% (n = 5) who underwent surgery, which was similar to our results summarized. Sixty-four percent (n = 9) of the patients who underwent complementary local treatments achieved a 12-month PFS. Goldner et al. performed a retrospective review of the National Cancer Database with an aim to define the role of metastasectomy in metastatic SCCA (2004–2014) [7]. Patients who underwent metastasectomy (n = 165) were compared to those who did not have metastasectomy (n = 2093). The authors stratified for liver-only metastases and found that the median OS time was increased for those who underwent liver resection (34 months vs. 16 months, *p* < 0.001). Omichi et al. reported retrospectively on 28 patients with metastatic SCC of the anus (n = 19), cervix (n = 2), tonsil (n = 2), lung (n = 2), unknown primary (n = 2), and vulva (n = 1) (1998–2015) [8]. The authors reported on outcomes after liver metastasis resection, with the five-year OS being 47%. Pawlik et al. reported on SCC patients who received liver-directed surgery for liver metastases [34]. Of the 52 patients evaluated, 27 patients had anal cancer. The median time to recurrence was 9.8 months, revealing a substantial time off systemic therapy. The authors identified that reduced DFS was seen in those patients with large hepatic metastases (>5 cm), positive surgical margins, and those with synchronous disease. Our analysis of only eight patients makes it difficult to determine any risks. 

Limitations are present in our review, as our sample size is small and has a short median follow-up period of 18 months. Of note, our institution sees approximately 150–200 new anal cancer patients per year, and the fact that the patients enrolled in multicenter metastatic trials are small highlights the rarity of these cases. Our review is retrospective and single-center in design; however, there are key findings identified in our analysis that show optimism for this strategy in select patients with disease that is biologically responsive to systemic therapy.

## 5. Conclusions

Based on the supporting literature and our current investigation, a subset of hepatic-only metastatic SCCA patients can benefit from hepatic metastasectomy with durable response and survival. The key will be determining the optimal patient selection for the greatest benefit. SPARTANA, a phase 2 trial, is currently recruiting [14]. SPARTANA is incorporating a phase 3 study portion that is evaluating multimodal treatment in oligometastatic anal cancer (ablative treatment, surgery, hypofractionated radiotherapy, radiofrequency ablation, or CRT). We support similar trial designs for future prospective investigation in this orphan disease, as we recognize that, given the rarity of the recruitment of the population for separate studies, exploring novel targets and the sequencing of multimodality treatment may not be feasible.

## Figures and Tables

**Figure 1 cancers-15-03890-f001:**
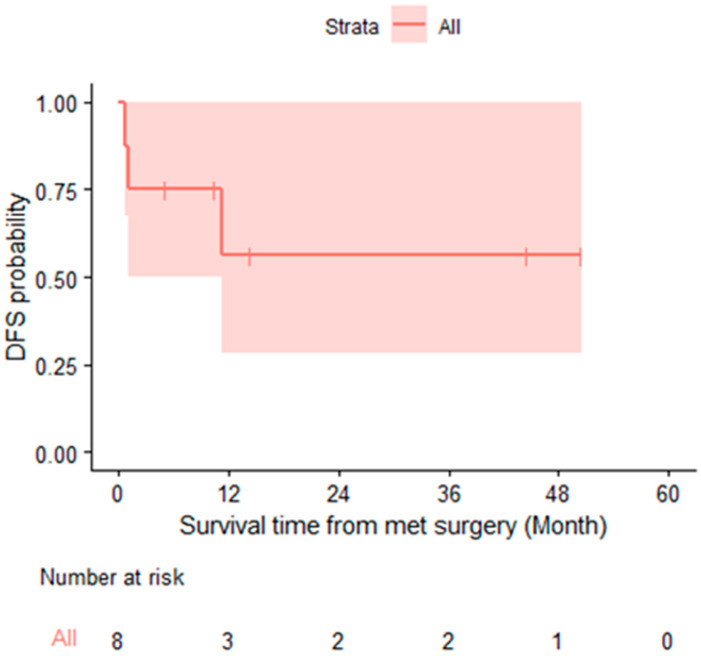
Disease-free survival probability.

**Table 1 cancers-15-03890-t001:** Baseline Demographics.

Characteristics	N (%)
Age	Median = 62 years old Range = 43–74 years old
Mutation profile DCLRE1C SNV, NOTCH1 del, TP53 SNV FBXW7, KMT2D, PIK3CA PIK3CA No profile	1 (12%) 1 (12%) 1 (12%) 5 (62%)
Metastatic disease diagnosis Metachronous Synchronous	4 (50%) 4 (50%)
Race Caucasian	8 (100%)
Gender Male Female	1 (12%) 7 (88%)
Prior Definitive CRT Yes No	8 (100%) 0 (0%)
CRT Radiation Sensitizer Chemotherapy Regimen 5-FU + Mitomycin C 5-FU + Cisplatin	4 (50%) 4 (50%)
ECOG Performance Status 0 1	6 (75%) 2 (25%)
Line of metastatic systemic therapy 1	8 (100%)
Systemic Therapy Regimen Clinical Trial Carboplatin + Paclitaxel 5-FU + Cisplatin	1 (12%) 5 (62%) 2 (25%)
Best Radiographic Response Any response Stable Progression	8 (100%) 0 0

CRT: chemoradiation; 5-FU: 5-Fluorouracil; ECOG: Easter Cooperative Oncology Group.

**Table 2 cancers-15-03890-t002:** Outcomes per Case.

Case	Systemic Therapy Provided	Duration of Systemic Therapy Prior to Hepatectomy (Months)	Hepatic Disease at Diagnosis	Hepatectomy	Pathologic Response	Recurrent Disease	Time to Recurrence or Last Follow-Up
**1**	Cisplatin + 5-FU	4	Multiple bilobar disease Largest lesion 11 cm	Partial segment 8 hepatectomy Segment 4 resection Segment 2 hepatectomy	pCR	No	54.9 months
**2**	Cisplatin + 5-FU	5	Solitary 9 × 5 cm left lesion	Segment 2 partial hepatectomy Segment 3 partial hepatectomy	pCR	No	48.5 months
**3**	Carboplatin + Paclitaxel	3	Multiple bilobar disease Largest lesion 1.3 cm	Segment 6 resection Segment 5 resection Segment 4 resection Segment 8 resection	pCR	No	15.7 months
**4**	Carboplatin + Paclitaxel	3	Solitary 10.3 × 5.6 cm right lesion	Right partial hepatectomy	5% tumor viability	No	5.7 months
**5**	Trial	2	Bilobar disease 3.6 × 5.4 cm left lesion 4 × 3.4 cm right hepatic lesion	Segment 8 resection Segment 2 resection Segment 3 resection	pCR	Yes Common hepatic artery lymphadenopathy	13.4 months
**6**	Carboplatin + Paclitaxel	3	Two isolated lesions 4.2 cm in segment 8 2 cm in segment 5	Segment 8 partial hepatectomy Segment 5 partial hepatectomy	40% tumor viability larger tumor; 100% viability in smaller tumor	Yes Right liver lesion	1.25 months
**7**	Carboplatin + Paclitaxel	3	Multiple lesions 1.5 cm in segment 5, 0.7 cm in segment 6, 0.4 cm in segment 1, 6 cm in segment 8	Segment 8 partial hepatectomy Segment 4 partial hepatectomy	pCR	No	11.5 months
**8**	Carboplatin + Paclitaxel	5	Multiple lesions 2 cm lesion in segment 4 1.5 cm and 0.9 cm in segment 2	Left hepatectomy	90% tumor viability	Yes Segment 5 1.2 cm	0.9 months

5-FU: 5-Fluorouracil; pCR: pathologic complete response.

## Data Availability

We will not be sharing data.

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
