# Peer review of "Hepatic Metastasectomy in Squamous Cell Carcinoma of the Anal Canal: A Case Series of a Curative Approach"

_cancers, 2023, doi:10.3390/cancers15153890_

Round 1

Reviewer 1 Report

The authors present a very interesting paper describing management and treatment of 8 patients with Squamous cell carcinoma of the anal canal with distant metastasis limited to the liver. 

As the authors recognize, the number of cases included in serie is small, and the conclusions that may be obtained are limited. The clinical problem studied in not frequent, so the paper is pertinent and informative. All cases have been thoroughly studied, and the 8 clinical cases patients are nicely described.

Only two minor questions: 1/ It would be interesting to know how many patients had been atended during this period with SCC anal carcinoma, and how many with metastatic disease at any localization. It will give idea of the dimension of the clinical problem that is studied in the paper.

2/ The follow-up period is short (median 18 months). This is a great limitation to obtain information of recurrences and survival. It would be important to underline this in the discussion.  

Reviewer 2 Report

I would like to congratulate the authors on their fascinating work regarding this interesting article on Hepatic metastasectomy in squamous cell carcinoma of the anal canal: a case series of a curative approach. Despite the major advances in colorectal surgical oncology, there are still numerous unanswered questions regarding which is the ideal management of patients with metastatic squamous cell carcinoma of the anal canal. The manuscript is well-written and the incorporated figure and tables make the study easy to follow.

1) This study includes the experience of eight hepatic-only metastatic squamous cell carcinoma of the anal canal cases who underwent hepatic resection. These patients underwent surgical resections after neoadjuvant systemic therapy.  The weak points of this study:

-the sample size is small

-retrospective study

-single-center experience

-only patients with hepatic metastases were included

What about patients with metastases in other organs? (number of patients, their treatment, compared with only liver metastases)

2) I would suggest a brief discussion on perineal colostomy, which can be preferred as surgery for these patients.

Consider citing the recently published article on this type of surgery with its advantages and disadvantages:

https://pubmed.ncbi.nlm.nih.gov/35664027/

Round 2

Reviewer 2 Report

The majority of requested changes were performed. It can be accepted for publication without further changes.